# Is There a Correlation between Dietary and Blood Cholesterol? Evidence from Epidemiological Data and Clinical Interventions

**DOI:** 10.3390/nu14102168

**Published:** 2022-05-23

**Authors:** Maria Luz Fernandez, Ana Gabriela Murillo

**Affiliations:** 1Department of Nutritional Sciences, University of Connecticut, Storrs, CT 06268, USA; 2Department of Biochemistry, University of Costa Rica, San Jose 11501-2060, Costa Rica; anagabriela.murillo@ucr.ac.cr

**Keywords:** dietary cholesterol, plasma cholesterol, lipoproteins, epidemiological studies, clinical interventions

## Abstract

Dietary cholesterol has been a topic of debate since the 1960s when the first dietary guidelines that limited cholesterol intake to no more than 300 mg/day were set. These recommendations were followed for several years, and it was not until the late 1990s when they were finally challenged by the newer information derived from epidemiological studies and meta-analysis, which confirmed the lack of correlation between dietary and blood cholesterol. Further, dietary interventions in which challenges of cholesterol intake were evaluated in diverse populations not only confirmed these findings but also reported beneficial effects on plasma lipoprotein subfractions and size as well as increases in HDL cholesterol and in the functionality of HDL. In this review, we evaluate the evidence from recent epidemiological analysis and meta-analysis as well as clinical trials to have a better understanding of the lack of correlation between dietary and blood cholesterol.

## 1. Introduction

The relationship between dietary and blood cholesterol is very controversial and has been debated within the scientific community since the 1960s when the first guidelines for dietary cholesterol were published [1]. It was not until the 2015 dietary guidelines that the upper limits of dietary cholesterol were eliminated, based on more current information [2]. The guidelines from the 1960s were not based on epidemiological data, meta-analysis or clinical intervention; they were construed as a consensus based on the current information available at that time [1]. Animal studies that have been used to study effects of dietary cholesterol on atherosclerosis, oxidative stress, and inflammation are not reliable since the concentration of cholesterol used varied from 1.25 to 5% of the diet [3,4,5], which would be the equivalent of approximately 10,000 to 37,000 mg/day. These dietary challenges with these exorbitant amounts of dietary cholesterol cannot possibly have any clinical application. Since those early years, the information regarding dietary cholesterol in humans has been substantially increased by key findings from epidemiological data of large cohort studies including the Framingham study as early as the 1980s [6], the Nurses’ study [7], National Health and Nutrition Examination Survey (NHANES) [8], and more recently others such as the Hellenic National Nutrition and Health Survey (HNNHS) [9]. In addition, clinical interventions in diverse populations including children [10], young adults [11,12], elderly people [13,14], obese individuals [15,16] metabolic syndrome populations [17,18], and diabetic patients [19,20] have demonstrated that the plasma biomarkers of coronary heart disease are not increased by dietary cholesterol (provided by eggs) but may result in the formation of less atherogenic lipoproteins [21,22].

There were two main objectives of this review: (1) To evaluate the most recent epidemiological evidence and meta-analysis that continue to support the lack of correlation between dietary and blood cholesterol and (2) to evaluate the effects of dietary cholesterol on plasma lipids and the atherogenicity of lipoproteins in recent clinical trials utilizing a dietary cholesterol challenge.

## 2. Epidemiolocal Evidence

Most studies that evaluate dietary cholesterol use eggs as a natural food source. The outdated dietary guidelines established a limit of 300 mg/day. Eggs contain approximately 180–200 mg of cholesterol in the yolk and have been identified as high cholesterol foods therefore they have been used to evaluate the effects of dietary cholesterol on blood cholesterol [8,23] However, observational and prospective studies have not found for the most part a direct relationship between egg consumption and blood cholesterol or cardiovascular disease (CVD) risk [24]. Although there are studies that have found correlations between dietary cholesterol, plasma cholesterol and heart disease risk [25,26].

Using data from the Prospective Urban Rural Epidemiology (PURE) study, Dehghan et al. evaluated egg consumption and CVD of individuals from 21 countries in a 9 year follow up [23]. The results showed that the higher egg intake (≥7 egg/week compared with <1 egg/week) was not significantly associated with blood lipids (including total cholesterol, HDL cholesterol, LDL cholesterol, total cholesterol/HDL cholesterol ratio, triglycerides, apolipoprotein (apo)A1, apo B, and apo B/apoA1 ratio, total mortality, or major CVD. Similar results were obtained with the data of other prospective studies: The ONTARGET (Ongoing Telmisartan alone and in Combination with Ramipril Global End Point Trial) and TRANSCEND (Telmisartan Randomized Assessment Study in ACEI Intolerant Subjects with Cardiovascular Disease). The three studies combined had more than 177,555 participants [23].

High blood pressure is considered a major risk factor for developing cardiovascular disease (CVD) [27]. Therefore, hypertensive patients must take further precautions to maintain a healthy diet, including, according to some researchers, reducing cholesterol intake. In a study conducted with data from the China Health and Nutrition Survey (CHNS) from 1991 to 2015, Wu et al. [28] evaluated the relationship between cholesterol intake from eggs and other sources and mortality among hypertensive patients. The study included 8095 participants who were followed-up for a mean of 11.4 years. Using regression analysis, the results indicated that people who consumed more than seven eggs per week had up to 29% lower mortality compared with patients who did not consume more than two eggs/week. Although egg consumption was inversely associated with mortality, cholesterol intake from other sources showed a positive association, suggesting that eggs may offer a protective effect that is not found in other cholesterol sources such as red meat, pork, cheese or butter, which also have a substantial amount of saturated fat, a nutrient positively linked to CVD [28,29]. Several authors have attributed this protective effect to eggs’ anti-inflammatory properties [30,31] and vitamin, mineral, and antioxidant content [8,32,33].

A recent large-scale cross-sectional study examined the relationship between dietary cholesterol and dyslipidemia in 8358 Chinese adults [34]. A total of 2429 participants (29% of the sample) were diagnosed with dyslipidemia according to the 2016 Chinese adult dyslipidemia prevention guide. Among all examinees, the mean cholesterol intake was 213.7 mg/day. According to the linear regression analysis, higher cholesterol consumption was associated with lower plasma TG and higher HDL-cholesterol in women (before adjusting for nutrient components), and no associations were observed in men. When the analysis was divided by cholesterol sources, eggs but no other sources showed a significant inverse relationship with the risk of dyslipidemia. In general, the results of this study showed a null association between cholesterol intake and serum lipids concentrations, indicating that consumption of cholesterol rich foods do not increase the risk of altering blood lipids and therefore, developing CVD [34]. Another epidemiological study including individuals (*n* = 3558) from the Hellenic National and Nutrition Health Survey (HNNHS) concluded that the more frequent egg consumption decreased the odds of dyslipidemia compared to either no eggs or infrequent egg intake [9] supporting the lack of effect of dietary cholesterol on plasma lipids and lipoproteins.

Table 1 shows the main epidemiological studies proving the lack of correlation between dietary cholesterol and blood cholesterol.

## 3. Meta-Analysis

A recent meta-analysis conducted by Godos et al. [35] revised the data of 39 prospective cohort studies that evaluated the association between egg consumption and the risk of CVD, coronary heart disease (CHD), and stroke. The studies analyzed included patients from North America, Europe, Asia, and some multi-national cohorts. In relation with CVD incidence and/or mortality (14 studies), the analysis showed that the intake of up to six eggs (a vehicle of dietary cholesterol) per week had an inverse association with CVD events, when compared to no intake. A similar trend was observed for CHD incidence and mortality (16 studies), where the risk decreased when examinees had up to two eggs per week. No associations between egg intake and stroke were found in this study, however a positive and linear association was found with egg consumption and heart failure. Still, the authors concluded that there is no evidence that eggs play a role in the development of CVD [35].

Berger et al. [36] analyzed 40 studies published between 1979 and 2013 for their meta-analysis. The authors included cohorts with participants without CVD diagnosis, either healthy or with risk CVD factors such as high blood pressure, dyslipidemia, diabetes, or metabolic syndrome present at baseline. Clinical interventions that recruited healthy individuals (with no CVD risk factors and no lipid lowering drugs use) at the beginning of the trial were also included in the analysis. The results showed no association between dietary cholesterol and coronary artery disease (CAD), ischemic stroke, or hemorrhagic stroke. However, cholesterol intake did affect blood lipids by increasing both serum total cholesterol and LDL cholesterol, although changes in LDL-C were not statistically significant when the intervention intake was excessive (>900/day). It is important to mention that HDL cholesterol also was also significantly increased by dietary cholesterol, which means no net change in CVD risk [36]. Other blood lipids such as plasma triglycerides remained unaffected. According to the authors, one of the reasons why some studies have found a positive linear association between cholesterol consumption and CVD outcomes is that many interventions do not control for other nutrients that affect that risk, for example saturated fatty acids and calories from fat, which can increase with cholesterol source intake and are positively correlated with CVD risk; on the contrary, dietary fiber and vegetable protein show a negative association with CVD and in some dietary patterns, like the Western diet, these two nutrients are inversely correlated with cholesterol intake [37,38].

Drouin-Chartier et al. [39] reviewed three large cohort studies: the Nurses’ Health Study (NHS) (1980–2012), NHS II (1991–2017) and The Health Professional Follow up Study (HPFS) (1986–2016) with the objective of establishing if there was an association between egg intake and the risk of developing CVD. In total, 14,806 subjects had diagnosed CVD (non-fatal myocardial infarction, fatal coronary heart disease or stroke) after the follow up. The analysis showed that most people consumed between one to five eggs per week and participants with a higher egg intake also had a higher BMI and were less likely to be under statin treatment. In this study, an increase of one egg per day was not associated with any CVD risk. In fact, in an updated meta-analysis that included multiple cohorts from the US, Europe, and Asia, moderate egg consumption was associated with no risk of developing CVD overall and lower risk in Asian populations [39].

Although is well stablished that dyslipidemia is a major CVD risk factor, altered blood lipids can also increase the risk for developing other chronic not transmittable diseases, such as type 2 diabetes mellitus (T2DM) [40,41]. In that matter, Drouin-Chartier et al. [42] also evaluated the association between egg intake and the risk of developing T2DM. Across the three large cohorts, higher egg intake was associated with lower prevalence of hypercholesterolemia, but 1 egg/day increase was associated with a 14% higher T2DM risk. However, in a random-effects meta-analysis of 16 prospective cohort studies (6 American, 8 European, and 2 Asian), no significant association between egg consumption and T2DM risk was found. It is noteworthy that there was a significant geographical heterogenicity in the results. In fact, among US studies, for each egg per day, T2DM risk increased by 18% but this positive association was not found in the studies conducted in Europe or Asia. This is hypothesized to be due to the relationship between eggs and other foods consumed with them in different regions. For example, in the three large cohorts mentioned before, egg intake was positively associated with total calories consumed, red meat, bacon and processed red meats, refined grains, potatoes, full-fat milk, and coffee, which are reflective of the Western diet. As previously stated, there is a large body of evidence that links this dietary pattern to obesity and other chronic diseases, including CVD and T2DM [42,43], so that association may be due for egg consumption pattern and not eggs or cholesterol alone. This suggest that further studies that control for all possible dietary cofounders or that investigate the reasons for such geographical differences are needed. The main meta-analysis showing the lack of correlation between dietary cholesterol and blood cholesterol is presented in Table 1.

**Table 1 nutrients-14-02168-t001:** Results from recent epidemiological studies and meta-analysis showing the lack of correlation between dietary cholesterol and blood cholesterol.

Population/Number of Studies	Association Assessed	Main Result	Reference (Year)
177,555 adults from PURE, TRASCEND and ONTARGET studies	Egg consumption with blood lipids and CVD	Higher egg intake is not associated with TC, LDL, TG, HDL, total mortality, or CVD.	[23] (2020)
8095 hypertense adults from the China Health and Nutrition Survey	Cholesterol intake from eggs and other sources and mortality	Cholesterol from eggs but not other sources is associated with lower mortality.	[28] (2020)
8358 Chinese adults	Dietary cholesterol and dyslipidemia	Cholesterol intake is associated with lower plasma TG and higher HDL-cholesterol in women, but not men. Cholesterol from eggs is associated with lower risk of dyslipidemia.	[34] (2022)
Three large cohort studies: NHS (1980–2012), NHS II (1991–2017) and HPFS (1986–2016). 16 prospective cohort studies (6 American, 8 European, and 2 Asian)	Egg intake and CVD risk	An increase of one egg per day is not associated with any CVD risk. Egg intake is associated with lower CVD risk in Asian populations.	[39] (2017)
39 prospective cohort studies from North America, Europe, and Asia	Egg consumption and the risk of CVD, CHD, and stroke	Consumption of six eggs per week has an inverse association with CVD events (but not stroke), when compared to no intake. No association is found for stroke.	[35] (2021)
40 studies with participants without diagnosed CVD		No association between dietary cholesterol and coronary artery disease (CAD), ischemic stroke, or hemorrhagic stroke. Dietary cholesterol increases total blood cholesterol, without affecting LDL/HDL ratio.	[36] (2015)
NHS (1980–2012), NHS II (1991–2017) and HPFS (1986–2016). 16 prospective cohort studies (6 American, 8 European, and 2 Asian)	Eegg intake and the risk of developing T2DM	Higher egg intake is associated with lower prevalence of hypercholesterolemia.	[42] (2020)

CVD: cardiovascular disease; CHD: coronary heart disease; TC: total cholesterol; LDL: low-density lipoprotein; HDL: high-density lipoprotein; TG: Triglyceride; NHS: the Nurses’ Health Study; HPFS: the Health Professional Follow up Study.

## 4. Clinical Interventions. Effects of Dietary Cholesterol on Plasma Lipids and Lipoprotein Subfractions

Clinical interventions across the life cycle have been conducted in children, young adults, elderly individuals [10,11,12,13,14], and those with conditions that put them at risk for chronic disease including obesity [15,16] and metabolic syndrome [17,18] to evaluate the effects of cholesterol challenges on plasma lipid concentrations and lipoprotein metabolism. Clinical interventions have also been conducted in people with type-2 diabetes [19,20]. The dietary cholesterol challenges have varied from adding between 200 mg/day of dietary cholesterol to 800 mg/day [10,11,12,13,14,15,16,17,18,19,20]. In all these interventions, eggs have been used as the vehicle for dietary cholesterol. In most of these studies, increases in HDL cholesterol have been observed [10,11,12,13,14,15,16,17,19]. In some studies, increases in LDL cholesterol have been reported following the cholesterol challenge [10,11,13,44]. However, the LDL/HDL ratio a very well-known biomarker for heart disease risk has either been maintained or has been decreased [10,11,12,13,14,15,16,17,18,19,20].

In a study following a protocol of 14 weeks in which young individuals consumed zero eggs for 2 weeks as the baseline, followed by one egg for 4 weeks, two eggs for 4 weeks and three eggs for the last 4 weeks, LDL cholesterol was lower or similar to baseline values during the whole intervention [12]. In contrast HDL cholesterol was higher than baseline all through the study from the intake of one to three eggs [12]. Other studies involving weight loss interventions [17], older people [14], overweight/obese individuals [15,16], and people with metabolic syndrome [17,18] that were challenged with two to three eggs per day (360–540 additional mg of dietary cholesterol) for extended periods of time, no increases in LDL cholesterol were observed, indicating that some individuals do have the ability to maintain LDL plasma cholesterol concentrations independent of the dietary cholesterol challenge.

Substituting high carbohydrate breakfast with eggs resulted in a similar lowering in plasma LDL cholesterol compared to baseline in 30 subjects following a randomized crossover study in which each breakfast was followed for 4 weeks [45]. However, lipoprotein subfractions were not different between dietary treatments [45].

Results from two randomized controlled studies conclude that the lack of effect of consuming 75 g or 150 g of eggs when compared to a no-egg diet on plasma LDL cholesterol was due to poor absorption of cholesterol from eggs [46]. Authors came to this conclusion by measuring the total cholesterol areas under the curve (AUC) between 0 and 10 hours [46]. Other studies have also shown no changes in LDL cholesterol and increases in HDL cholesterol after 56 individuals consumed one egg/day for 12 weeks [47] or in 12 sedentary young adults undergoing endurance exercise for 8 weeks [48]. In contrast, another study where subjects consumed either zero eggs (*n* = 34 per group) or two eggs per day five times a week for 14 weeks, no changes were observed in HDL cholesterol but also LDL cholesterol was not affected [49].

Regarding lipoprotein metabolism, dietary cholesterol leads to the formation of the large LDL particles that are known to be less atherogenic [32] and reduces the concentration of small LDL [10,16], which has been recognized as a highly atherogenic particle for its ability to become oxidized, penetrate the arterial wall and initiate the atherosclerosis process [50]. In the study conducted in Latino children who have higher concentrations of the small LDL, a shift toward larger LDL diameter and less of the small LDL subfractions was observed [10]. Increases in the large LDL after intake of 640 mg/d of cholesterol (3 eggs) resulted in higher concentrations of large LDL compared to 0 mg of additional cholesterol in young individuals [51,52], elderly subjects [53] and in obese individuals [54]. Increases in large LDL have also been noticed in young individual consuming up to 640 mg cholesterol per day [12]. Women and men who participated in a randomized clinical trial in which they consumed either three eggs or the equivalent amount of egg substitute, presented increases in the large buoyant less atherogenic LDL during the whole egg period [55].

Small HDL size has been correlated atherogenic dyslipidemia [56] highlighting the importance of increases in HDL size and in the number of large HDL particles following egg consumption. These increases in large HDL have been observed after a cholesterol challenge in the elderly [56]. Comparable results have been shown in young and healthy individuals [51,52] and in those with metabolic syndrome [57]. Interestingly, increases in the large HDL as well as compositional changes resulted in an HDL particle with increased cholesterol efflux capacity [22,58], which is also a better transporter for the carotenoids lutein and zeaxanthin in plasma [56,57], A summary of studies showing changes in lipoprotein subfractions and size following a cholesterol challenge is presented in Table 2.

## 5. Mechanisms to Manage Dietary Cholesterol

The epidemiological data and the clinical interventions presented above clearly indicate the lack of correlation between dietary and blood cholesterol. These observations also suggest that the body has specific mechanisms to manage excesses of dietary cholesterol. The proposed mechanisms including decreased absorption or synthesis suppression were recognized early on [59] and they explain why dietary cholesterol cannot be extrapolated directly to plasma cholesterol. The absorption of dietary cholesterol varies according to each individual and it comprises cholesterol from food, biliary cholesterol, and to a certain extent intestinal epithelial sloughing [60]. The transport of cholesterol to the liver involves several steps including solubilization in micelles, transport to the enterocytes, incorporation into chylomicrons, and transport through lymph and blood vessels to the liver and other tissues. The absorption of cholesterol varies from 29 to 80% with an average of 60% [61]. Niemann-Pick C1-Like-1 (NPCL1), a receptor localized in the intestinal cells, has a major role in cholesterol absorption [61].

In terms of synthesis, intracellular cholesterol exerts a negative feedback regulation on HMG-CoA, the rate limiting enzyme of cholesterol synthesis. Excess cholesterol in cells leads to the suppression of HMG-CoA reductase activity [62]. HMG-CoA reductase can be regulated by changes in phosphorylation but most importantly by changes in transcription [63]. The transcriptional regulation involves the binding of sterol response element binding protein (SREBP) to sterol response elements located in the 5′ region of the CoA reductase gene [64]. This is also an important mechanism by which individuals maintain plasma cholesterol levels and can consume high amounts of dietary cholesterol and do not increase plasma concentrations. One clear case study that exemplifies the compensatory mechanisms to maintain cholesterol homeostasis is the case of the 88-year-old individual who consumed 25 eggs per day (about 4500 mg/day) and had no heart problems and normal plasma cholesterol levels [65]. The compensatory mechanisms were a marked reduction in cholesterol absorption, increased synthesis of bile acids, and reduced cholesterol synthesis [65].

Therefore, the handling of dietary cholesterol by the body can be explained by decreased absorption and down-regulation of synthesis. Figure 1 depicts the mechanisms by which the body handles dietary cholesterol and maintains plasma cholesterol homeostasis.

## 6. Conclusions

We confirm from the review of the literature on epidemiological data, meta-analysis, and clinical interventions where dietary cholesterol challenges were utilized that there is not a direct correlation between cholesterol intake and blood cholesterol. This lack of correlation is mainly due to the compensatory mechanisms exerted by the organism to manage excess dietary cholesterol, including decreases in cholesterol absorption and down-regulation of cholesterol synthesis. A great number of epidemiological studies and meta-analysis indicate that dietary cholesterol is not associated with CVD risk nor with elevated plasma cholesterol concentrations. Clinical interventions in the last 20 years demonstrate that challenges with dietary cholesterol do not increase the biomarkers associated with heart disease risk. Further, in the specific circumstances where eggs are the source of dietary cholesterol, an improvement in dyslipidemias is observed due to the formation of less atherogenic lipoproteins and changes in HDL associated with a more efficient reverse cholesterol transport. However, if the cholesterol sources are consumed with saturated and trans fats, as happens in the Western diet pattern, increases in plasma cholesterol may be observed. The most recent epidemiological data and clinical interventions for the most part continue to support the USDA 2015 dietary guidelines that removed the upper limit of dietary cholesterol.

## Figures and Tables

**Figure 1 nutrients-14-02168-f001:**
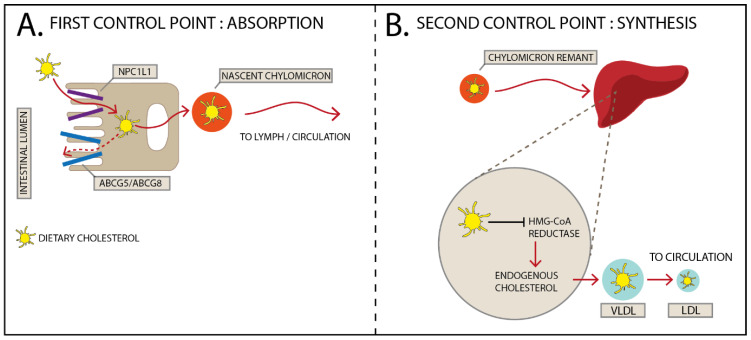
Mechanisms of how dietary cholesterol affects cholesterol metabolism. (**A**) Dietary cholesterol enters the enterocyte via NPC1L1 after being released from the micelle. However, some cholesterol is effluxed back to the intestinal lumen via ABCG5 and ABCG8 transporters explaining why only a percentage of the cholesterol consumed in the diet reaches the bloodstream. The remaining cholesterol gets packed into nascent chylomicrons, which enter the lymph and then the systemic circulation. (**B**) After losing most of its triglycerides, the cholesterol-loaded chylomicron remnant is removed by the liver. In the liver, free cholesterol inhibits HMG-CoA reductase, the rate limiting enzyme for endogenous cholesterol synthesis, Thus, if more cholesterol is consumed, less will be synthesized by the hepatocytes. VLDL: very-low-density lipoprotein; LDL: low-density lipoprotein; NPC1L1: Polytopic Niemann-Pick C1-like 1; ABCG5/ABCG8: ATP-binding cassette transporters G5/G8; HMG-CoA: 3-hydroxy-3-methyl glutaryl coenzyme A.

**Table 2 nutrients-14-02168-t002:** Beneficial Modifications in LDL and HDL size and subfractions due to dietary cholesterol.

Lipoprotein	Dietary Cholesterol Intake and Population	Changes	Reference (Year)
LDL Diameter Compared to added 0 mg/d cholesterol	510 mg/day for 4 weeks in children	LDL diameter was larger	[10] (2005)
Large LDL compared to 0 added mg/cholesterol	640 mg/day for 4 weeks in elderly people	Higher concentrations of large LDL	[53] (2006)
Large LDL compared to 0 mg of added dietary cholesterol	210, 425, and 640 mg/day in young individuals for 4 weeks each	Higher concentrations of large LDL	[51] (2017)
Large LDL Compared to an oatmeal breakfast	640 mg/day for 4 weeks in young population	Higher concentrations of large LDL	[52] (2018)
Large LDL: Compared to 0 mg of added dietary cholesterol	640 mg/day for 4 weeks in an overweight/obese population	Higher concentrations of large LDL	[54] (2010)
Small LDL: Compared to 0 mg of added dietary cholesterol	210, 425, and 640 mg/day in young individuals for 4 weeks each	Lower concentrations of small LDL	[51] (2017)
Small LDL: Compared to 0 mg of dietary cholesterol	640 mg/day for 4 weeks in an overweight/obese population	Lower concentrations of small LDL	[54] (2010)
HDL Diameter: Compared to 0 mg of added dietary cholesterol	640 mg/day for 4 weeks in elderly people	Larger HDL diameter	[53] (2006)
Large HDL: Compared to 0 mg of added dietary cholesterol	210, 425, and 640 mg/day in young individuals for 4 weeks each	Higher concentrations of large HDL	[51] (2005)
Large HDL: Compared to an oatmeal breakfast	640 mg/day for 4 weeks in young population	Higher concentrations of large HDL	[52] (2018)
Large HDL: Compared to 0 mg of added dietary cholesterol	640 mg/day for 4 weeks in an overweight/obese population	Higher concentrations of large HDL	[54] (2010)

## Data Availability

Not applicable.

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
