# Peer review of "Is There a Correlation between Dietary and Blood Cholesterol? Evidence from Epidemiological Data and Clinical Interventions"

_nutrients, 2022, doi:10.3390/nu14102168_

Round 1

Reviewer 1 Report

The article concludes through a literature review of epidemiological data, meta-analyses, and clinical interventions for dietary cholesterol challenges that there is no direct correlation between cholesterol intake and blood cholesterol. The lack of this correlation is mainly due to compensatory mechanisms imposed by the body to manage dietary cholesterol, including reduced cholesterol absorption and downregulation of cholesterol synthesis. The article has a clear logic and clear thinking, uses a wealth of literature to confirm that there is no direct correlation between cholesterol intake and blood cholesterol, and cites clinical intervention literature to make the conclusion more convincing. Our summarized comments are as follows:

  1. The literature reviewed confirms that there is no direct correlation between cholesterol intake and blood cholesterol. I think there is some chance, because factors such as dietary habits and climate environment in different countries and regions may have some influence on cholesterol.

  1. The references cited at the conclusion of the article are too rich, please put forward your own thoughts.

  1. It is recommended to have more literature on experiments to confirm your conclusions.

  1. Why only eggs are used as the research standard? Cholesterol-rich foods There are many articles mentioning red meat, pork, etc. but lacking relevant descriptions.

  1. There are some syntax errors. Please recheck the manuscript;

  1. Regarding the discussion of the relevant content of the article, please pay attention to the logic of the content.

Author Response

The article concludes through a literature review of epidemiological data, meta-analyses, and clinical interventions for dietary cholesterol challenges that there is no direct correlation between cholesterol intake and blood cholesterol. The lack of this correlation is mainly due to compensatory mechanisms imposed by the body to manage dietary cholesterol, including reduced cholesterol absorption and downregulation of cholesterol synthesis. The article has a clear logic and clear thinking, uses a wealth of literature to confirm that there is no direct correlation between cholesterol intake and blood cholesterol, and cites clinical intervention literature to make the conclusion more convincing. Our summarized comments are as follows:

We thank the reviewer for all the comments that helped us improve the content of this manuscript

  1. The literature reviewed confirms that there is no direct correlation between cholesterol intake and blood cholesterol. I think there is some chance, because factors such as dietary habits and climate environment in different countries and regions may have some influence on cholesterol.

  1. That is an interesting point raised by the reviewer and there is a relationship when dietary cholesterol is taken in combination with saturated fat as we explain in our paper (lines 131-136). We also mention geographic differences in our paper (lines 151-156).
  2. The references cited at the conclusion of the article are too rich, please put forward your own thoughts.
  3. We have modified the conclusions as suggested by the reviewer

 It is recommended to have more literature on experiments to confirm your conclusions.

We have extended our search and have added 9 more articles

  1. Why only eggs are used as the research standard? Cholesterol-rich foods There are many articles mentioning red meat, pork, etc. but lacking relevant descriptions.

Those articles on other cholesterol-containing foods are rich in saturated fat, so this is a huge confounding variable. This is why, all studies that want to test cholesterol utilize eggs since they are rich in dietary cholesterol, but saturated fat is low.

  1. There are some syntax errors. Please recheck the manuscript;

We have carefully revised the manuscript and corrected all syntax errors, especially those found in the conclusion. Thanks for checking

 Regarding the discussion of the relevant content of the article, please pay attention to the logic of the content.

  1. We have paid close attention to the logic of the content. We expect that this has improved the readability of the manuscript

This is the logic of the article

  1. In the introduction we discuss the progress in the dietary guidelines that removed the upper limit of dietary cholesterol and now we are including the two main objecrtives that were missing and we thank the reviewer for pointing this out: 1. A review of the more current epidemiological data that still support the 2015 dietary guidelines 2. Effects of dietary cholesterol on plasma lipids and atherogenicity of lipoproteins
  2. We then proceed to discuss the epidemiological evidence from recent years that support the lack of correlation between dietary cholesterol, blood cholesterol and risk for heart disease.
  3. Next we discuss recent meta-analysis on the same topic
  4. The next component is the clinical studies and effects of dietary cholesterol on plasma lipids and lipoproteins
  5. We follow with an explanation regarding why this lack of correlation between dietary and blood cholesterol
  6. Finally we have a discussion to tie this all together.

Reviewer 2 Report

The topic of the manuscript may be interesting, but it needs more improvement due to methodological shortcomings.

I feel confused reading this manuscript. This has no clear structure. It would be helpful to clarify the nature of the review. In order for the manuscript to report significant scientific achievements, the principles of systematic review writing should be adopted. Currently, the manuscript does not meet these rules. It should meet the PRISMA criteria (PRISMA criteria are described e.g. here doi: 10.1136/bmj.n160).

In its current form, the purpose is unclear. What is the main thesis? What do the authors think the term "recent epidemiological analysis, meta-analysis and clinical trials" means? Is it a collection and discussion of publications that have emerged between 2015 and the present and relating this knowledge to current recommendations? This needs to be clearly defined. What were the criteria for including papers in the review? What bibliographic databases were used by the authors? How many articles were reviewed? What were the keywords on the basis of which bibliographic databases were searched?

Table 1. and 2. are poorly described. Make these tables more detailed. Also add the year of publication.

The sentence in lines 115-119 is too long and thus not very readable.

It appears that the text in lines 239-247 is a figure caption, but it is in regular font. This is unclear.

What is missing is a concise discussion that summarizes the results and relates them to current recommendations.

Lines 262-264, the conclusions reached in the manuscript may support previous recommendations, but not vice versa.

Author Response

I feel confused reading this manuscript. This has no clear structure. It would be helpful to clarify the nature of the review. In order for the manuscript to report significant scientific achievements, the principles of systematic review writing should be adopted. Currently, the manuscript does not meet these rules. It should meet the PRISMA criteria (PRISMA criteria are described e.g. here doi: 10.1136/bmj.n160).

  1. Thanks for providing this information related to Systematic Reviews. I would like to make a clarification.

There are three types of review articles: Reviews, Systematic Reviews and Meta-Analysis. Our current paper is a is a review article on the relationship between dietary cholesterol and blood cholesterol based on current epidemiological data and clinical studies where plasma lipids and more importantly lipoprotein subfractions were measured.

It was never meant to be a systematic review.  If this had been a systematic review, this should have been included in the title

Nutrients publishes those 3 types of articles, I would really appreciate it if  the reviewer evaluates this manuscript asa Review not as a Systematic Revies

This is the logic of the article

  1. In the introduction we discuss the progress in the dietary guidelines that removed the upper limit of dietary cholesterol and now we are including the two main objecrtives that were missing and we thank the reviewer for pointing this out: 1. A review of the more current epidemiological data that still support the 2015 dietary guidelines 2. Effects of dietary cholesterol on plasma lipids and atherogenicity of lipoproteins
  2. We then proceed to discuss the epidemiological evidence from recent years that support the lack of correlation between dietary cholesterol, blood cholesterol and risk for heart disease.
  3. Next we discuss recent meta-analysis on the same topic
  4. The next component is the clinical studies and effects of dietary cholesterol on plasma lipids and lipoproteins
  5. We follow with an explanation regarding why this lack of correlation between dietary and blood cholesterol
  6. Finally we have a discussion to tie this all together. Again, we thank the reviewr for allowing us to see our shortcomings so that the paper is more understandable

In its current form, the purpose is unclear. What is the main thesis? What do the authors think the term "recent epidemiological analysis, meta-analysis and clinical trials" means? Is it a collection and discussion of publications that have emerged between 2015 and the present and relating this knowledge to current recommendations? This needs to be clearly defined. What were the criteria for including papers in the review? What bibliographic databases were used by the authors? How many articles were reviewed? What were the keywords on the basis of which bibliographic databases were searched?

We have defined the main objectives now in the Introduction to make it clear for the reviewer. This was never intended to be a systematic review. We are addressing 2 questions that are now clearly presented in the introduction. We have added more references although we found very few papers addressing the alterations of lipoproteins due to egg consumption.

Table 1. and 2. are poorly described. Make these tables more detailed. Also add the year of publication.

We respectfully disagree with the reviewer regarding Table 1 since it contains all pertinent information. Table 2 has been redone by adding more detailed information about the change in lipoproteins. The year of the publications are now included in both Tables as suggested by the reviewer

The sentence in lines 115-119 is too long and thus not very readable.

The reviewer is right. We have separated it into 2 sentences

It appears that the text in lines 239-247 is a figure caption, but it is in regular font. This is unclear.

The Figure caption has been redone to improve clarity. Thanks for pointing out this problem. I have seen that the figure caption has the same font size as the rest of the manuscript in published papers. This is something that should be corrected during the galley proofs by editorial office if this is not the case if the paper is accepted

What is missing is a concise discussion that summarizes the results and relates them to current recommendations.

We agree with the reviewer we have modified the conclusion substantially

Lines 262-264, the conclusions reached in the manuscript may support previous recommendations, but not vice versa.

The reviewer is right. We have changed the phrase to make it more clear

Round 2

Reviewer 2 Report

I appreciate the changes made by the authors. However, there is still some inadequacy when it comes to providing a clear time range of the literature collected and the criteria for inclusion in the review. This makes the paper unclear and gives rise to speculation that the reporting of studies may be biased. I do not think that the authors found it difficult to describe the search strategy and inclusion criteria.